# Accelerating Quadratic Optimization with Reinforcement Learning

**Jeffrey Ichnowski**[*1], **Paras Jain**[*1], **Bartolomeo Stellato**[2], **Goran Banjac**[3], **Michael Luo**[1],
**Francesco Borrelli**[1], **Joseph E. Gonzalez**[1], **Ion Stoica**[1], and **Ken Goldberg**[1]
[1]University of California, Berkeley,   [2]Princeton University,   [3]ETH Zürich
Correspondence to: {`jeffi, paras_jain`}@berkeley.edu
[*]equal contribution

## Abstract

First-order methods for quadratic optimization such as OSQP are widely used for large-scale machine learning and embedded optimal control, where many related problems must be rapidly solved. These methods face two persistent challenges: manual hyperparameter tuning and convergence time to high-accuracy solutions. To address these, we explore how Reinforcement Learning (RL) can learn a policy to tune parameters to accelerate convergence. In experiments with well-known QP benchmarks we find that our RL policy, RLQP, significantly outperforms state-of-the-art QP solvers by up to 3x. RLQP generalizes surprisingly well to previously unseen problems with varying dimension and structure from different applications, including the QPLIB, Netlib LP and Maros-Mészáros problems. Code, models, and videos are available at `https://berkeleyautomation.github.io/rlqp/`.

## 1  Introduction

Solving quadratic programs (QPs) efficiently is critical to applications in finance, robotic control and operations research. While state-of-the-art interior-point methods scale poorly with problem dimensions, first-order methods for solving QPs typically require thousands of iterations. Moreover, real-time control applications have tight latency constraints for solvers [33]. Therefore, it is important to develop efficient heuristics to solve QPs in fewer iterations.

The Alternating Direction Method of Multipliers (ADMM) [6, 15, 18] is an efficient first-order optimization algorithm, and is the basis for the widely used and state-of-the art Operator-Splitting QP (OSQP) solver [44]. ADMM performs a linear solve on a matrix based on the optimality conditions of the QP to generate a step direction, and then projects the step onto the constraint bounds.

While state-of-the-art, the ADMM algorithm has numerous hyperparameters that must be tuned with heuristics to regularize and control optimization. Most importantly, the step size parameter $\rho$ has considerable impact on the convergence rate. However, is still unclear how to select $\rho$ before attempting the QP solution. While some theoretical works compute the optimal $\rho$ [17], they rely on solving semidefinite optimization problems which are much harder than solving the QP itself. Alternatively, some heuristics introduce "feedback" by adapting $\rho$ throughout optimization in order to balance primal and dual residuals [44, 6, 22].

We propose RLQP (see Fig. 1), an accelerated QP solver based on OSQP that uses reinforcement learning (RL) to adapt the internal parameters of the ADMM algorithm between iterations to minimize solve times. An RL algorithm learns a policy $\pi_\theta \colon \mathcal{S} \to \mathcal{A}$, parameterized by $\theta$ (e.g., the weights of a neural network), that maps states in a set $\mathcal{S}$ to actions in set $\mathcal{A}$ such that the selected action maximizes the accumulated reward $r$. To train the policy for RLQP, we define $\mathcal{S}$ to be the internal state of the

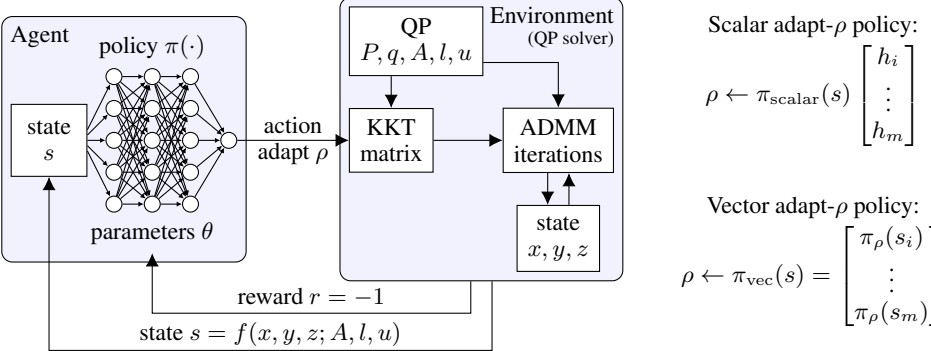

Figure 1: RLQP uses deep reinforcement learning (RL) to compute a policy that adapts the internal parameters of a first-order quadratic program (QP) solver to speed up the solver's convergence rate. In a standard RL formulation, a policy computes an action based on its observation of the state of the environment, and taking the action results in a change in state and a reward. In RLQP, the policy is parameterized by a neural network, the state is a function of the solver's internal state parameterized by the QP, the action changes a parameter ($\rho$) of the solver, and the reward minimizes QP solve time. RLQP proposes 2 formulations of RL and associated policy: scalar and vector.

QP solver (e.g., the constraint bounds, the primal and dual estimates), $\mathcal{A}$ to be the adaptation to the internal parameter ($\rho$) vector, and $r$ to minimize the number of ADMM iterations taken.

RLQP's policy can be trained either jointly across general classes of QPs or with respect to a specific class. The general version of RLQP is trained once on a broad class of QPs and can be used out-of-the-box on new problems. The specialized version of RLQP is trained on a specific class of problems that the solver will repeatedly encounter. While this requires additional setup and training time, it is useful when QPs will be repeatedly solved in application (e.g., in a 100 Hz control loop).

In experiments, we train RLQP on a set of randomized QPs, and compare convergence rates of RLQP to non-adaptive and heuristic adaptive policies. To compare generalization and specialization, we investigate RLQP's performance in the settings where 1) the train and test sets of QPs come from the same class of problems, 2) the train set contains problems from a superset of classes contained in the test set, 3) the train set contains a subset, and 4) when the train and test sets are from distinct classes. In the results section we show that RLQP outperforms OSQP by up to 3x.

The contributions of this paper are:

- Two RL formulations to train policies that provide coarse (scalar) and fine (vector) grain updates to the internal parameters of a QP solver for faster convergence times
- Policies trained jointly across QP problem classes or to specialize to specific classes
- Experimental results showing that RLQP reduces convergence times by up to 3x and generalizes to different problem classes and outperform existing methods

## 2  Related Work

This work touches a number of related research areas, including convex optimization, ML-accelerated optimization, learning in 1st-order methods and RL. We summarize related work in Table 1.

**Convex optimization**   Many researchers have proposed algorithms for quadratic programs, which generally fall into three classes: active set [47], interior point [37], and first-order methods. Of the active set and interior point solvers, perhaps the most well-known are Gurobi [20] and MOSEK [36]. Active-set solvers operate by iteratively adapting an active set of constraints based on the cost function gradient and dual variables [39]. Interior-point solvers iteratively introduce and vary barrier functions to represent constraints and solve unconstrained convex problems. We instead base this work on a first-order method solver, OSQP [44]. One of the advantages of OSQP over interior points solvers, is that it can readily be warm started from a near-by solution, as is common in many applications such as solving a sequential quadratic program [41] and solving QPs for model-predictive control.

**ML-accelerated combinatorial optimization**   Accelerating combinatorial optimization problems with deep learning has been explored with wide application [4, 5], including branch-and-bound for

|  | Continuous state space | Accelerate specialized 1$^{\text{st}}$-order problems | Guaranteed convergence | Domain agnostic | Scalable to arbitrary # of dims. | Generalize to novel problems |
|---|---|---|---|---|---|---|
| Khalil et al. [27] | ✗ | ✗ | ✓ | ✓ | ✓ | ✓ |
| Dai et al. [9] | ✗ | ✗ | ✗ | ✗ | ✓ | ✗ |
| Li and Malik [28] | ✓ | ✓ | ✗ | ✓ | ✗ | ✗ |
| Metz et al. [34] | ✓ | ✓ | ✗ | ✓ | ✓ | ✓ |
| Wei et al. [46] | ✓ | ✓ | ✓ | ✗ | ✗ | ✗ |
| RLQP | ✓ | ✓ | ✓ | ✓ | ✓ | ✓ |

Table 1: **Summary of related work.** RLQP meets all objectives while demonstrating state-of-the-art performance on challenging optimization problem sets.

mixed-integer linear programming [1, 27], graph algorithms [9] and boolean satisfiability problems (SAT) [7]. Many combinatorial optimization problems have exponential search spaces and are NP-hard in a general setting. However, learning-augmented combinatorial algorithms utilize very different methods to RLQP as combinatorial problems have discrete search spaces.

**Learning in first-order methods**  Accelerating first-order methods with machine learning has gained considerable recent interest. Li and Malik [28] demonstrate a learned optimization algorithm outperforms common first-order methods for several convex problems and a small non-convex problem. Metz et al. [34] show a learned policy outperforms first-order methods when optimizing neural networks, but finds that directly learning parameter update values can be sensitive to exploding gradient problems. We avoid this instability during optimization by learning a policy to adapt parameters of the ADMM algorithm. Wei et al. [46] recently proposed an RL agent to tune parameters for an ADMM-based inverse imaging solver.

**Reinforcement Learning Overview**  Reinforcement learning (RL) algorithms include both on-policy algorithms, such as Proximal Policy Optimization [42], REINFORCE [45], and IMPALA [11], and off-policy algorithms, such as DQN [35] and Soft Actor Critic [21]. RLQP extends the off-policy Twin-Delayed DDPG (TD3) [13], an actor-critic framework with an exploration policy for continuous action spaces that extends Deep Deterministic Policy Gradient (DDPG) algorithm [29] while addressing approximation errors. Furthermore, in one formulation of RLQP, we train a shared policy for multiple agents following an RL approach proposed by Huang et al. [23]. With this single policy, RLQP updates multiple parameters using state associated with each constraint of a QP.

## 3   Background

In this section, we summarize QPs, the OSQP solver, and an MDP formalization.

### 3.1   Quadratic Programs

A quadratic program with $n$ variables and $m$ constraints takes the form:

$$\begin{aligned}
\text{minimize} \quad & (1/2)x^T P x + q^T x \\
\text{subject to} \quad & l \leq Ax \leq u,
\end{aligned}$$

where $x \in \mathbb{R}^n$ is the optimization variable, $P$ is an $n \times n$ symmetric positive semi-definite matrix that defines the quadratic cost, $q \in \mathbb{R}^n$ defines the linear cost, $A$ is an $m \times n$ matrix that defines the $m$ linear constraints, and $l, u \in \mathbb{R}^m$ are the constraint's lower and upper bounds. Here, $\leq$ is an element-wise less-than-or-equal-to operator. In this form, to specify an equality constraint, the lower and upper bounds are set to the same value, and to specify a constraint unbounded from one side, a sufficiently large value (or $\pm\infty$) is specified for the other side.

### 3.2   First-Order QP Solver Algorithm

The solver we speed up is OSQP, which uses a first-order ADMM method to solve QPs. We summarize OSQP here. Given a QP, OSQP first forms a *KKT* matrix (below), then iteratively refines a solution from an initialization point for vectors $x^{(0)} \in \mathbb{R}^n$, $y^{(0)} \in \mathbb{R}^m$, and $z^{(0)} \in \mathbb{R}^m$, where the

superscript in parenthesis refers to the iteration. Each iteration computes the values for the $k+1$ iterates by solving following linear system (e.g., with an $LDL^T$ solver):

$$\underbrace{\begin{bmatrix} P + \sigma I & A^T \\ A & \mathrm{diag}(\rho)^{-1} \end{bmatrix}}_{\text{KKT matrix}} \begin{bmatrix} x^{(k+1)} \\ v^{(k+1)} \end{bmatrix} = \begin{bmatrix} \sigma x^{(k)} - q \\ z^{(k)} - \mathrm{diag}(\rho)^{-1} y^{(k)} \end{bmatrix} \tag{1}$$

and then performing the following updates:

$$\tilde{z}^{(k+1)} \leftarrow z^{(k)} + \mathrm{diag}(\rho)^{-1}(v^{(k+1)} - y^{(k)})$$

$$z^{(k+1)} \leftarrow \Pi_{[l,u]}\left(\tilde{z}^{(k+1)} + \mathrm{diag}(\rho)^{-1} y^{(k)}\right)$$

$$y^{(k+1)} \leftarrow y^{(k)} + \mathrm{diag}(\rho)\left(\tilde{z}^{(k+1)} - z^{(k+1)}\right),$$

where $\sigma \in \mathbb{R}_+$ and $\rho \in \mathbb{R}_+^m$ are regularization and step-size parameters, and $\Pi_{[l,u]} : \mathbb{R}^m \to \mathbb{R}^m$ projects its argument on the constraint bounds. We use the notation $\mathrm{diag} : \mathbb{R}^m \to \mathbb{S}^m$ to denote the operator that maps a vector to a diagonal matrix. We define the primal and the dual residual vectors as

$$\xi_{\text{primal}}^{(k)} = Ax^{(k)} - z^{(k)}, \quad \text{and} \quad \xi_{\text{dual}}^{(k)} = Px^{(k)} + q + A^T y^{(k)}.$$

When the primal and dual residual vectors are small enough in norm after $k$ iterations, $x^{(k+1)}$ and $y^{(k+1)}$ are primal and dual (approximate) solutions to the QP.

Internally, OSQP has a single scalar $\bar{\rho}$ that it uses to form $\rho$ according to the following formula:

$$\rho = \bar{\rho} \begin{bmatrix} h_1 \\ \vdots \\ h_m \end{bmatrix}, \text{ where } h_i = \begin{cases} 1 & \text{if } l_i \neq u_i \text{ (inequality constraints)} \\ 10^3 & \text{if } l_i = u_i \text{ (equality constraints)}, \end{cases} \tag{2}$$

where the subscript $i$ denotes the $i$-th coefficient of $h$, and the bounds $l$ and $u$.

Periodically, between ADMM iterations, OSQP will adapt the value of $\bar{\rho}$. The existing hand-crafted formula for adapting $\rho$ attempts to balance between primal and dual residuals, by setting $\bar{\rho}^{(k+1)} \leftarrow \bar{\rho}^{(k)} \sqrt{\|\xi_{\text{primal}}\|_\infty / \|\xi_{\text{dual}}\|_\infty}$. Empirically, adapting $\rho$ between iterations can speed up the convergence rate.

### 3.3   Multi-Agent Single-Policy MDP

In a Markov Decision Process (MDP), an *agent* can be in any state $s \in \mathcal{S}$, take an action $a \in \mathcal{A}$, and with the transition dynamics function, $\mathcal{T}(\cdot \mid s, a)$, transitions from state $s$ to state $s'$ after taking action $a$. The agent receives a reward $R: \mathcal{S} \times \mathcal{A} \to \mathbb{R}$ for transitioning from $s$ to $s'$ by taking action $a$. Given a tuple $(\mathcal{S}, \mathcal{A}, T, R, \gamma)$, the MDP optimization objective is to find a policy $\pi_\theta : \mathcal{S} \to \mathcal{A}$, parameterized by $\theta$, that maximizes the expected cumulative reward $\mathbb{E}\left[\sum_{t=0}^{\infty} \gamma^t r^t\right]$, where $r^t$ is the reward at time $t$ and $\gamma \in [0, 1)$ is a discount factor.

We also formulate a multi-agent single-policy MDP setting in which $m$ agents collaborate in a shared environment in state $s_{\text{env}} \in \mathcal{S}_{\text{env}}$. At each time step, each collaborating agent (CA) $i$ has its own state $s_i \in \mathcal{S}_{\text{ca}}$, action $a_i \in \mathcal{A}_{\text{ca}}$, and observations $o_i \in \mathcal{O}$, but, for computation feasibility, share a single policy $\pi_\theta : \mathcal{S}_{\text{ca}} \to \mathcal{A}_{\text{ca}}$. State transitions for the environment and all $m$ agents occur simultaneously according to a state transition function $\mathcal{T} : \mathcal{S}_{\text{env}} \times \mathcal{S}_{\text{ca}}^m \times \mathcal{A}_{\text{ca}}^m \to \mathcal{S}_{\text{env}} \times \mathcal{S}_{\text{ca}}^m$ and result in a single shared reward $R : \mathcal{S}_{\text{env}} \times \mathcal{S}_{\text{ca}}^m \times \mathcal{A}_{\text{ca}}^m \to \mathbb{R}$ and discount factor. The objective is to find a single shared policy $\pi_\theta$ that maximizes the expected cumulative reward. This can be thought of as a special case of a multi-agent MDP [31] or Markov game [30], and we adapt a formulation from Huang et al. [23].

## 4   Method

The goal of RLQP is to learn a policy to adapt the $\rho \in \mathbb{R}^m$ vector used in the ADMM update in (1) (see Fig. 1). As the dimensions of this vector vary between QPs, we propose two methods that can handle the variation in $m$. The first method learns a policy to adapt a scalar $\bar{\rho}$ and then applies (2) to populate the coefficients of the $\rho$ vector. The second method learns a policy to adapt individual coefficients of the $\rho$ vector.

| **Algorithm 1** TD3 for $\bar{\rho}$ (scalar) | **Algorithm 2** TD3 for $\rho$ (vector) |
|---|---|
| 1: **Input:** exploration noise $\sigma$, buffer size `rs` | 1: **Input:** exploration noise $\sigma$, buffer size `rs` |
| 2: $\pi, Q \leftarrow$ initialize policy and critic (see TD3) | 2: $\pi, Q \leftarrow$ initialize policy and critic (see TD3) |
| 3: $\mathcal{D} \leftarrow$ replay buffer w/ `rs` | 3: $\mathcal{D} \leftarrow$ replay buffer w/ `rs`×(avg no. of constraints) |
| 4: `env`, $s^{(0)} \leftarrow$ new QP, state | 4: `env`, $\mathbf{s}^{(0)}, m \leftarrow$ new QP, state, no. of constraints |
| 5: **for** $t \in \{0, \dots, T\}$ **do** | 5: **for** $t \in \{0, \dots, T\}$ **do** |
| 6: $\quad \bar{\rho}^{(t)} \leftarrow \pi(s^{(t)}) + \epsilon, \ \epsilon \sim \mathcal{N}(0, \sigma)$ | 6: $\quad \rho_i^{(t)} \leftarrow \pi(s_i^{(t)}) + \epsilon, \ \epsilon \sim \mathcal{N}(0, \sigma) \ \forall i \in [1 \mathinner{..} m]$ |
| 7: $\quad s^{(t+1)}, r^{(t)}, \texttt{done}^{(t)} \leftarrow \texttt{step}(\texttt{env}, s^{(t)}, a^{(t)})$ | 7: $\quad \mathbf{s}^{(t+1)}, r^{(t)}, \texttt{done}^{(t)} \leftarrow \texttt{step}(\texttt{env}, \mathbf{s}^{(t)}, \boldsymbol{\rho}^{(t)})$ |
| 8: $\quad$ store $(s^{(t)}, \bar{\rho}^{(t)}, r^{(t)}, s^{(t+1)})$ in $\mathcal{D}$ | 8: $\quad$ store $(\mathcal{D}, s_i^{(t)}, \rho_i^{(t)}, r^{(t)}, s_i^{(t+1)}) \ \forall i \in [1 \mathinner{..} m]$ |
| 9: $\quad$ **if** $\texttt{done}^{(t)}$ **then** | 9: $\quad$ **if** $\texttt{done}^{(t)}$ **then** |
| 10: $\quad\quad$ `env`, $s^{(t)} \leftarrow$ new QP, state | 10: $\quad\quad$ `env`, $\mathbf{s}^{(t)}, m \leftarrow$ new QP, state, no. of constr. |
| 11: $\quad$ update $\pi$ and $Q$ using data sampled from $\mathcal{D}$ | 11: $\quad$ update $\pi$ and $Q$ using data sampled from $\mathcal{D}$ |

Since both the number of variables $n$ and the number of constraints $m$ can vary from problem to problem, and the same QP can be written in $(n!\, m!)$ permutations, we propose learning policies that are problem size and permutation invariant. To do this, we provide a permutation-invariant fixed-size state of the QP solver to either policy.

## 4.1 RL Policy for Scalar Adaptation

To speed up convergence of OSQP, we hypothesize that RL can learn a scalar $\bar{\rho}$ adaptation policy ($\pi_{\text{scalar}}$) that can perform as-well-as or better than the current handcrafted policy ($\pi_{\text{hc}}$) of OSQP. The handcrafted policy in OSQP periodically adapts $\rho$ by computing a single scalar $\bar{\rho}$, then sets the coefficients of $\rho$ based on the value of $\bar{\rho}$. In both handcrafted and RL cases, the policy is a function $\pi : S_{\bar{\rho}} \to A_{\bar{\rho}}$, where $S_{\bar{\rho}} \in \mathbb{R}^2$ are the norms of the primal and dual residuals stacked into a vector, $A_{\bar{\rho}} \in \mathbb{R}$ is the value to set to $\bar{\rho}$. One advantage of this approach is that a simple heuristic can check that the proposed change to $\bar{\rho}$ is sufficiently small and leave the KKT matrix unchanged to avoid a costly matrix factorization.

To compute this policy, $\pi$, we use Twin-Delayed DDPG TD3 [13], an extension of deep-deterministic policy gradients (DDPG) [29], as the action space is continuous. We summarize TD3 in Alg. 1. TD3 learns the parameters $\theta$ of a policy $\pi_{\text{scalar}}$ network and critic $Q$ network, where $\pi_{\text{scalar}}$ determines the action to take and $Q_\pi(s, a) = r(s, a) + \gamma \mathbb{E}_{s' \sim \mathcal{T}(\cdot|s,a)}[Q_\pi(s', \pi(s'))]$ is the expected reward for a given state-action pair following the recursive Bellman equation. TD3 updates $Q_\pi$ by minimizing the loss on the Bellman equation, and updates the policy network using a policy gradient [45] of the objective

$$ J(\theta) = \mathbb{E}_{s \sim \mathcal{D}}[R(s, \pi_{\text{scalar}}(s))], $$

that is,

$$ \nabla_\theta J = \mathbb{E}_{s \sim \mathcal{D}}[\nabla_\theta \pi_{\text{scalar}}(s) \nabla_a Q_\pi(s, a)|_{a = \pi_{\text{scalar}}(s)}] $$

where $\mathcal{D}$ is the discounted state visitation distribution [43]. For brevity, we leave out some details of TD3 in the algorithms, including: $Q$ is composed of two networks, the minimum value of the two networks estimates the reward, exploration noise is clamped, and $\pi_{\text{scalar}}$ updates are staggered.

In RLQP, the "environment" `env` is an instance of a randomized QP problem, and a call to `step()` applies a change to $\bar{\rho}$ (and thus via (2) to $\rho$), advances a QP a fixed number of ADMM iterations, and returns the updated internal state $s$, a reward $r$, and a termination flag `done`. In this case, the internal state $s$ is a vector containing the current norms of the primal and dual residuals of the QP. The reward $r$ is $-1$ if not done, and $0$ if the QP is solved.

We train with randomized QPs across various problem classes (Sec. 5) that have solutions guaranteed by construction. To ensure progress, we set a step limit (not shown in the algorithm) since bad actions can cause the solver to fail to converge in a timely manner. During training, we also always adapt $\rho$ in each step and ignore the heuristic adapt/no-adapt policy.

For well-scaled QPs, the residuals and $\rho$ can reasonably range between $10^{-6}$ and $10^6$. Since this can cause issues with training the policy networks, we train the policy network with logs of the residuals, and exponentiate the network's output to get the action to apply.

## 4.2  RL Policy for Vector Coefficient Adaptation

For some classes of QPs, the solver can further speed up convergence by adapting all coefficients of the vector $\rho$, instead of applying (2) to a scalar $\bar{\rho}$. Conceptually, this could be accomplished with a policy $\pi_{\text{vec}} : S_{\text{QP}} \to A_{\text{vec}}$, where $S_{\text{QP}} \in \mathbb{R}^{O(n+m)}$ is the internal state of the solver and $A_{\text{vec}} \in \mathbb{R}^m_+$ is the new value for $\rho$. However, due to variation in problem size and permutation, we instead propose a simplification in which $\pi_{\text{vec}}$ is formulated as a policy $\pi_\rho : S_\rho \to A_\rho$ that is applied per coefficient of $\rho$. Here, $S_\rho \in \mathbb{R}^6$ is state corresponding to a single coefficient in $\rho$, and $A_\rho \in \mathbb{R}$ is the value to set for that coefficient.

To define $S_\rho$, we observe that coefficients in $\rho$ are one-to-one with coefficients in $y$, $z$, $l$, $u$, and $Ax$. We observe that constraint bounds are likely to have an impact on an ADMM iteration when coefficients of $z$ are "close" to their bounds in $l$ or $u$. A coefficient in $z$ is also "close" to a solution when it is nearly equal to the corresponding coefficient in $Ax$. Finally, to include a permutation-invariant signal on the overall convergence, we borrow the idea of max-pooling inputs from graph neural networks [40, 3] and include the infinity norms of the primal and dual residuals of the QP solver. We thus define a coefficient's state as:

$$s_i = \begin{bmatrix} \min(z_i - l_i, u_i - z_i) \\ z_i - (Ax)_i \\ y_i \\ \rho_i \\ \|\xi_{\text{primal}}\|_\infty \\ \|\xi_{\text{dual}}\|_\infty \end{bmatrix} \in S_\rho.$$

In practice, we clamp values in each state $s_i$ to reasonable ranges (e.g., $[10^{-8}, 10^6]$, $[-10^6, 10^6]$, $[-10^6, 10^6]$, $[10^{-6}, 10^6]$, $[10^{-6}, 10^6]$, $[10^{-6}, 10^6]$ for the coefficients of $s_i$, in order). Empirically, training is more efficient if the policy operates on states with the log of the first and last 3 coefficients.

Since each `step` in the vector formulation applies $m$ actions and updates $m$ states simultaneously, we adapt the multi-agent single-policy TD3 formulation from Huang et al. [23], and show it in Alg. 2, next to Alg. 1, with the main differences highlighted in blue. Before each step, `step` applies the policy with exploration noise to generate $m$ actions corresponding to $m$ coefficient updates to $\rho$. After each `step`, Alg. 2 adds the $m$ tuples to the replay buffer with each tuple consisting of coefficient-specific states and actions, along with the single *shared* reward for the step. Since each step results in $m$ tuples added to the replay buffer, Alg. 2 allocates a replay buffer large enough to hold the average number of tuples expected to be in the individual QPs in the training set.

The hypothesis of this approach is that the some coefficients, and thus policy actions for coefficients, will have more of an effect on convergence, and thus the reward, than others. When the domain for the policy function has more of an effect, the range of the actions will have lower variance. Similarly, when the policy values have less effect, the variance will be higher. This suggests that when training the policy network in this case, having a lower learning rate, and higher batch size can help. A lower learning rate will cause smaller gradient steps when training the network so that it does not overfit to some part of the high variance training data. A higher batch size will allow gradients to average out in high variance training data so that the gradient step better matches the true mean of the data.

## 5  Experiments

To train and test the proposed methods, we modify OSQP to support direct querying and modification of its $\rho$ vector, and integrate both $\pi_{\bar{\rho}}$ and $\pi_{\text{vec}}$ policies for benchmarking, and a runtime flag to switch between policies. We train the network using randomly generated QPs from OSQP's benchmark suite. The form of these QPs falls into 7 classes (see below), but the specific coefficient values in the objective and constraints are generated from a random-number generator. These QPs are also guaranteed to be feasible by construction (e.g., by reverse engineering constraint values from a pre-generated solution). To separate train and test sets, we ensure that each set is generated from uniquely seeded random-number generators. Training is performed in PyTorch with a Python wrapper around the modified OSQP which is written C/C++. During benchmarking, the solver performs runtime adaptation of $\rho$ using PyTorch's C++ API on the already-trained policy network. We train a small model to keep runtime network inference as fast as possible.

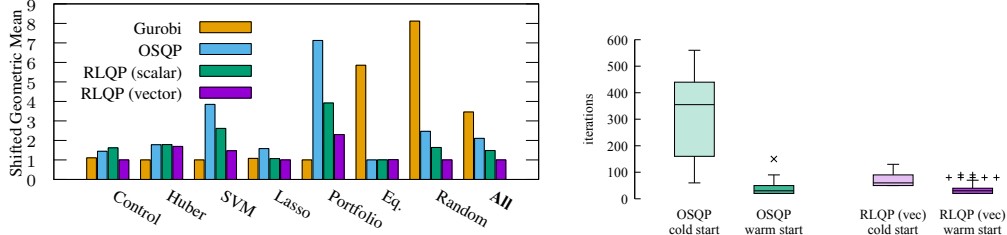

Figure 2: **Left:** Comparison of *general* adaptation policy applied to different classes. We train an RL policy using multiple classes, and show the performance per class, along with each class. The $y$-axis is the shifted geometric mean across problems within each class, and the value of 1 is always assigned to the best in class. The right-most **All** class is the aggregate of all classes to the left of it. **Right:** Comparison of warm-starting performance using OSQP's warm-start benchmark.

We evaluate all policies with 7 problem domains (referred to as the "benchmark problem") defined in Appendix A of the paper on OSQP [44]. These policies cover control, Huber fitting, support-vector machines (SVM), Lasso regression, Portfolio optimization, equality constrained, and random QP domains. Alongside RLQP, we benchmark the unmodified OSQP solver to evaluate how the RL policy improves convergence. For both RLQP and OSQP, we evaluate to the high-accuracy settings $(10^{-5})$ of the OSQP benchmarks. While our focus is on improving the first-order method in OSQP with an RL policy, we include some benchmarks against the state-of-the-art commercial Gurobi solver [20] (using default settings) as it may be of interest to a practitioner.

We consider three evaluation configurations: (1) *multi-task policy* learning in which we train a single RLQP policy on a suite of random benchmark problems and test it across all problems, (2) *class-specific policy* learning in which we train and test the policy for a single problem domain and (3) *zero-shot generalization* where we test a general policy on a novel unseen problem class.

We evaluate speedups with the shifted geometric mean [19] as problems have wide variations in runtime across several orders of magnitude. This metric is the standard benchmark used by optimization community. The shifted geometric mean is computed as:

$$\exp \sum_{i=1}^{N} (1/N) \log(\max(1, v_i + s)) - s,$$

where $v_i$ is compute time in seconds, $s = 10$, and $N$ is the number of values (e.g., QPs solved).

We also evaluate on QPLIB [14], Netlib [16], and Maros and Mészáros [32], as they are well-established benchmark problems in the optimization community.

In all experiments, the policy network architecture has 3 fully-connected hidden layers of 48 with ReLU activations between the input and output layers. The input layer is normalized, and the output activation is Tanh. The critic network architectures use the identity function as the output activation, but otherwise matches the policy. As small networks for fast CPU inferences are desirable here, we attempted to keep the network as small as possible. We performed minimal experimentation before settling on this architecture—finding that smaller networks fail to converge during training.

We trained on a system with 256 GiB RAM, two Intel Xeon E5-2650 v4 CPUs @ 2.20 GHz for a total of 24 cores (48 hyperthreads), and five NVIDIA Tesla V100s. We ran benchmarks on a system with Intel i9 8-core CPU @ 2.4 GHz and without GPU acceleration.

## 5.1 Multi-task/General RLQP Policy

We train a general policy on a broad set of problem classes and compare solve times with different classes. During training, we sample one of seven QP domains from benchmark problem. From that sampled problem domain, we generate a random problem.

In Fig. 2, we compare the shifted geometric mean of solving 10 problems of 20 different dimension, for a total of 200 runs per class per solver. The problem dimensions for Control, Huber, SVM, Lasso are (10, 11, 12, 13, 14, 16, 17, 20, 23, 26, 31, 37, 45, 55, 68, 84, 105, 132, 166, 209); for Random

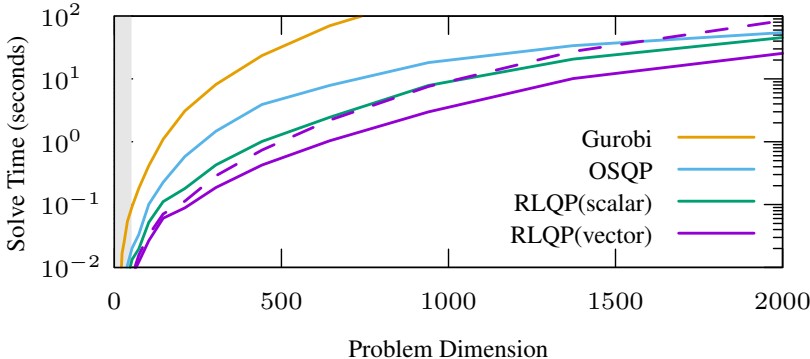

Figure 3: Solve time with increasing dimension on the Random QP problem set. We train and benchmark two vector RL adaptation policies: (dashed) on problems ranging from dimension 10 to 50, and (solid) on problems ranging from 10 to 2000. The gray box shows the range of the training data for the dashed line. When the benchmark run is in the same problem-dimension distribution as the training data, the relative performance between solvers is consistent, however, when the problem dimension is outside of he training distribution, the performance diverges.

and Eq are (10, 11, 12, 13, 15, 18, 23, 29, 39, 53, 73, 103, 146, 211, 304, 442, 644, 940, 1373, 2009), and for Portfolio are (5, 6, 7, 8, 9, 10, 12, 14, 16, 20, 24, 28, 35, 43, 52, 65, 80, 99, 124, 154). From the results, we observe that both RLQP adaptation policies typically improve upon convergence rate from the handcrafted policy in OSQP, and in some cases, e.g. Portfolio optimization, by up to 3x.

## 5.2 Problem Dimension Scaling

To test how a trained policy scales to higher dimensions, we train a policy on low dimensional problems (10 to 50), and solve problems with varying dimensions, including dimensions higher than the training set (up to 2000). For comparison, we also include a policy trained on the full dimension range (10 to 2000). From the results plotted in Fig. 3, we observe that a policy trained on a lower dimensional training set, can show improvement beyond its training range. However, as the problem size diverges more from the training set, its performance suffers and it eventually loses to the handcrafted policy. Both low-dimensional and full-dimension range polices, were trained using the same network architecture, we hypothesize that this behavior is a function of the training data and not a limitation of the network expressiveness. While this is a disadvantage of using smaller problems for training, in practice it may be outweighed by the advantage in training time—as each RL step requires $O((n + m)^3)$ compute time.

## 5.3 Training a Class-Specific Policy

Many applications in control [25] and optimization [26] require QPs from the same class to be repeatedly solved. To test if training a policy specific to a QP class can outperform a policy trained on the benchmark suite, we train policies specific to the problems generated by the trust-region [8] based solver for sequential quadratic program (SQP) from a grasp-optimized motion planner (GOMP) [25, 24] for robots. With these problems, RLQP trained on the benchmarks converges more slowly than the handcrafted policy included in OSQP. With a vector policy trained on the the QPs from the SQP, the shifted geometric mean of OSQP is 1.37. This result suggests that while a general policy may work for multiple problem classes, there are cases in which it is beneficial to train a policy specific to a problem class, particularly if the QPs from that problem class are repeatedly solved.

## 5.4 Warm Starting QPs

One benefit of first-order method such as OSQP is their ability to warm start—that is, rapidly converge from a good initial guess. We test if RLQP retains the benefit of warm start on OSQP's warm-start benchmark and show the results in Fig. 2 (right). As warm starts require fewer iterations, and thus

| Inst. | $n$ | $m$ | non-zeros | OSQP | RLQP (scalar) | RLQP (vector) |
|---|---|---|---|---|---|---|
| 8845 | 1546 | 777 | 10999 | 6.386 | timeout | **5.435** |
| 9002 | 2890 | 1649 | 12580 | **6.000** | timeout | timeout |
| 8906 | 5223 | 838 | 20781 | 1.108 | 1.447 | **0.741** |
| 8559 | 10000 | 5000 | 24998 | 59.648 | 205.372 | **24.083** |
| 8938 | 4001 | 11999 | 31997 | timeout | timeout | **0.991** |
| 8567 | 10000 | 7500 | 32497 | 98.511 | 284.112 | **22.222** |
| 8616 | 13870 | 10404 | 41610 | 0.126 | **0.113** | 0.141 |
| 8515 | 16002 | 8002 | . 56005 | **0.105** | timeout | timeout |
| 8785 | 10399 | 11362 | 63023 | 6.334 | timeout | **2.972** |
| 8495 | 27543 | 8000 | 73029 | 1.612 | 0.742 | **1.174** |
| 8602 | 34552 | 52983 | 242887 | 99.872 | timeout | **55.629** |
| 8547 | 1003001 | 1001000 | 6003001 | timeout | timeout | timeout |

Table 2: **QPLIB problems**. Timing results for solving the convex continuous QPs with constraints from QPLIB [14]. The Inst. column is QPLIB's instance number. The columns $n$ (number of variables), $m$ (number of constraints), and non-zeros indicate the QP's complexity. A *timeout* result indicates the solver terminated due to reaching an iteration or time limit (300 s). We hypothesize that the RLQP timeouts are due to out of distribution test problems, as the policy here was trained on the benchmark classes.

fewer adaptations than cold starts, we expect RLQP to show a smaller improvement here. In the plot, we can see that RLQP retains the benefit of warm starting, and also gains a improvement over OSQP.

## 5.5 QPLIB

We benchmark convex continuous QP instances with constraints from QPLIB [14], and show the results in Table 2. Since there are only a few such QPLIB instances and they come from varying classes, creating a train/test split is problematic. We thus use the general policy trained on the benchmark classes. From the table, we observe that the general RLQP policy beats OSQP's heuristic policy in all but three cases. In two cases RLQP fails due to reaching an iteration or time limit. Training on similar problems should help avoid a timeout.

## 5.6 Netlib Linear Programming benchmark

The Netlib Linear Programming benchmark [16] contains 98 challenging real-world problems including supply-chain optimization, scheduling and control problems. As with the QPLIB benchmark, we evaluate results with a general policy trained on the benchmark classes. We solve problems to high-accuracy as many of these benchmarks are poorly scaled. Overall, vector formulation of RLQP is $1.30\times$ faster than OSQP by the scaled geomean of runtimes. We include a problem-specific breakdown in the supplementary materials.

## 5.7 Maros and Mészáros

In a manner similar to the QPLIB problems, we also benchmark on the Maros and Mészáros repository of QPs. [32]. This collection of 138 QP problems, includes many poorly scaled problems that cause OSQP to fail to converge. We compute the shifted geometric mean for problems solved by both OSQP and RLQP with the general vector policy. RLQP converges faster, with OSQP's shifted geometric mean is 1.829 times that of RLQP. Because the dataset contains 138 problems, a table of the full results is included in the Supplementary Material.

## 5.8 Learned Policy Analysis

We plot and evaluate a learned policy in Fig. 4. From the plot, we see that when $z$ approaches either bound, the output $\rho$ increases. We also observe that with higher $y$, the policy outputs a higher $\rho$. Both of these observations are consistent with the intent and observations behind OSQP's heuristic policy: a high $\rho$ is desirable when the primal residual is smaller than the dual, and a low $\rho$ is desirable

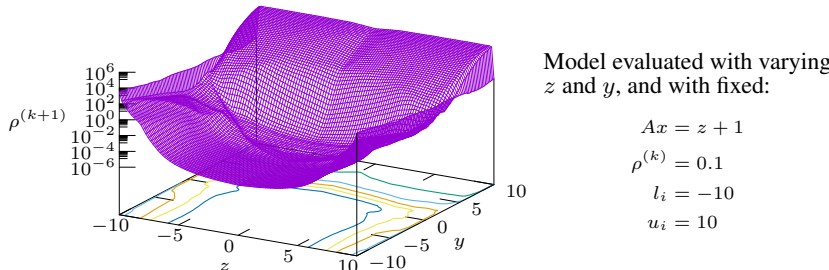

Figure 4: **Trained model for general vector policy,** evaluated with varying $z$ and $y$, and fixed $Ax$, $\rho^{(k)}, l_i, u_i$. As $z$ approaches either bound and with increasing $y$, the policy outputs higher $\rho$.

when the dual is smaller than the primal. However, the learned policy is not a simple square root of residuals like OSQP. The results suggest that the optimal policy is non-linear and problem specific.

## 6  Limitations

RLQP has limitations. For QPs that converge after few iterations, and thus do not adapt $\rho$, having a better adaptation policy is moot. Training RLQP can take a prohibitively long time and require a large replay buffer for some applications, for example, to train the benchmark suite of QPs required several days on a high-end computer with 256 GiB—this may be mitigated to an extent by sharing learned policies between interested practitioners. The time it takes to evaluate the RL policy, especially the vector version, may reduce the performance benefit of faster convergence—this may be mitigated by learning more efficient representations, or by using dedicated neural-network processing hardware.

## 7  Conclusion

We presented RLQP, a method for using reinforcement learning (RL) to speed up the convergence rate of a first-order method quadratic program solver. RLQP uses RL to learn a policy to adapt the internal parameters of the solver to allow for fewer iterations and faster convergence. In experiments, we trained a generic policy and results suggest that a single policy can improve convergence rates for a broad class of problems. Results for a problem-specific policy suggest that fine-tuning can further accelerate convergence rates.

In future work, we will explore whether additional RL policy options can speed up convergence rate further, such as training a hierarchical policy [2] in which the higher-level policy determines the interval between adaptation, performing a neural-architecture search [10], using meta-learning [12, 38] to speed up problem-specific training, and online-learning to adjust the policy at runtime to adapt to changing problems. Furthermore, we will investigate how to analyze the learned policies (e.g., Fig. 4) to devise new optimization algorithms with analytical step-size updates and faster theoretical convergence rates.

## Acknowledgements

This research was performed at the AUTOLAB at UC Berkeley in affiliation with the Berkeley AI Research (BAIR) Lab, and the CITRIS "People and Robots" (CPAR) Initiative. In addition to NSF CISE Expeditions Award CCF-1730628, this research is supported by gifts from Amazon Web Services, Ant Group, Ericsson, Facebook, Futurewei, Google, Intel, Microsoft, Nvidia, Scotiabank, Splunk and VMware. Any opinions, findings, and conclusions or recommendations expressed in this material are those of the authors and do not necessarily reflect the views of the sponsors. We thank Ashwin Balakrishna, Arnav Gulati as well as other colleagues for their helpful feedback.

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
