# Accelerating Quadratic Optimization with Reinforcement Learning
# ※ Supplemental Material ※

**Jeffrey Ichnowski**[*1]**, Paras Jain**[*1]**, Bartolomeo Stellato**[2]**, Goran Banjac**[3]**, Michael Luo**[1]**,**
**Francesco Borrelli**[1]**, Joseph E. Gonzalez**[1]**, Ion Stoica**[1]**, and Ken Goldberg**[1]
[1]University of California, Berkeley,   [2]Princeton University,   [3]ETH Zürich
Correspondence to: {jeffi, paras_jain}@berkeley.edu
[*]equal contribution

## A    Implementation

Training the scalar policy for OSQP [7] requires no modification of the OSQP source code. Instead, we disable the builtin `adaptive_rho` setting and set `max_iter` and `check_termination` to the interval to associate with the policy (e.g., 100). With these settings, the solver will run for the preset iteration count and either return "solved" or "iteration limit reached." Upon reaching the iteration limit, the RL policy step applies the adaptation via an existing call. On the subsequent step, the internal state of the QP solver remains otherwise unchanged, thus this process mimics adapting the $\rho$ in the inner loop of he solver.

Training the vector policy requires a minor modification of OSQP to support setting and getting the internal $\rho$ vector. Otherwise, training the vector policy is the same as training the scalar policy.

Using and benchmarking the policy requires additional modification of the solver. We modify the code so that when the `adaptive_rho` setting is enabled, OSQP calls through the PyTorch C++ API [6] to pass the internal state through the learned policy network and then apply the adaptation internally.

We parallelize the training implementation to run multiple episodes concurrently, but otherwise follow close to the TD3 [2] algorithm for the scalar policy, and according to the one-policy [4] modifications described in the main text. When training reaches an update or epoch step, the implementation waits for concurrently running episodes to complete before updating the networks—this leads to imprecise step counts between training, but does not appear to otherwise effect training.

We plot the training curves on learning the benchmark problems in Fig. 1. In this figure we observe that the policy and critic loss lowers over training time, and correspondingly that the episode length (which is the negative reward), goes down as the learned policy improves.

## B    Comparison and Ablation of Training and Policies

We compare multiple training runs with different seeds for different model architectures, and plot the results in Fig. 2. The *Vector 1* policy does not include residuals $\xi_{\mathrm{primal}}$ and $\xi_{\mathrm{dual}}$ in $S$, while *Vector 2* and *Vector 3* policies do. The *Vector 1* and *Vector 2* policies are networks with 3 hidden layers, while *Vector 3* has 2 hidden layers, all layers are 48 wide with ReLU activations. All policies were trained for a maximum of 50 epochs, with a replay buffer size of $4 \times 10^8$, $10^5$ initial steps, updates every 10000 steps, 5000 batch size, 20000 steps per epoch, 0.995 polyak, 1.0 noise, 2.5 noise clip, and policy updates every other critic update. For 3-layer networks, we set the learning rate to $10^{-5}$ for both policy and critic networks, and for the 2-layer network, we set the learning rate to $10^{-6}$. We selected the epoch with the lowest average loss, though better performance may be possible with a

35th Conference on Neural Information Processing Systems (NeurIPS 2021).

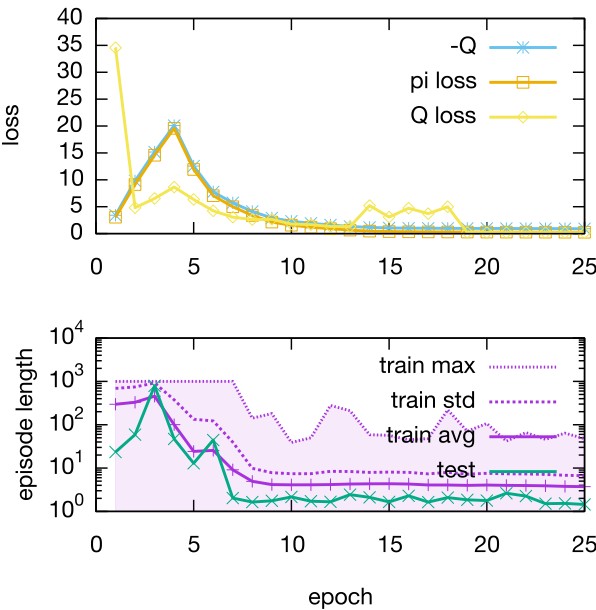

Figure 1: **Reinforcement learning training curves.** In these plots, we show the training curves over a training run. The top graph shows the policy (pi) and critic (Q) loss, along with the negated average critic (-Q) value. The bottom graph shows the training episode length maximum (train max), average length + standard deviation (train std), and average length (train avg), and the test episode average. The top graph converges to smaller loss indicating that the policy and critic are improving. The bottom graph shows that average and maximum episode length lowers as training continues.

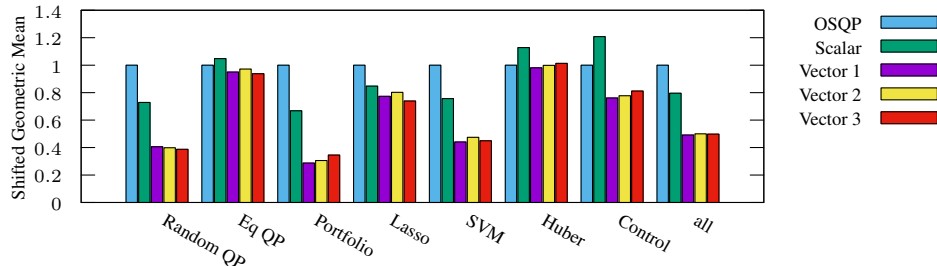

Figure 2: Comparison of the geometric mean of solve times for policies from different training runs. Here we normalize to the geometric mean of OSQP at 1.0. See text for description of the policies and how they were trained.

policy from a different epoch. We observe minor variation in the 3 trained policies, but not sufficient to categorically state which one is the best.

## C   Netlib Linear Programming Results

In order to measure how well the vector RL policy for OSQP generalizes to unseen inputs, we evaluate the policy on the 98 Netlib LP test problems [3]. These problems are a collection of linear programs considered to be large and challenging. We select this benchmark as this class of linear programs is significantly different than any of the quadratic program classes we train with.

Overall, the vector RLQP policy outperforms the OSQP policy with a shifted geometric mean runtime that is $1.30\times$ faster. Moreover, the vector RLQP policy solves 5.2% more problems than the heuristic OSQP. Figure 3 shows the number of problems solved by OSQP and RLQP with increasing runtime.

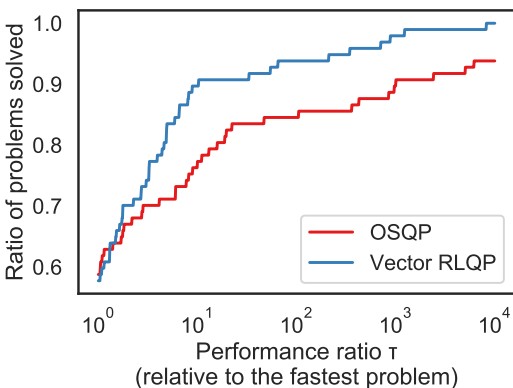

Figure 3: **Netlib LP performance profiles** We evaluate how the learned RLQP policy generalizes to unseen problems. The vector policy is $1.3\times$ faster (shifted geomean) than the existing heuristic in OSQP while solving 5.2% more problems.

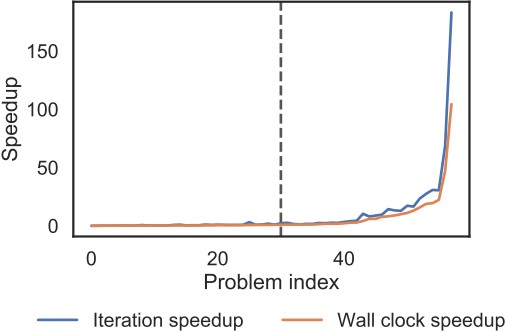

Figure 4: **Netlib LP problem speedup** Iteration speedup per problem in the Netlib LP problem set. Problems right of the dotted line observe speedup greater than 1. For the majority of problems, RLQP accelerates convergence by up to $73\times$.

Performance ratio ($\tau$) represents the rescaled runtime relative to the fastest problem, following the practice of Dolan and Moré [1].

These results are slightly better than the Netlib LP results included in the main paper. With the extra time, we were able to slightly tune the training procedure. Namely, we reduced the replay buffer size (which avoids training the policy with stale rollouts), decreased the learning rate, increased the batch size and finally trained the policy longer. These changes do not substantially change results (from $1.23\times$ to $1.30\times$). Moreover, the Netlib LP problems require a large number of iterations from the OSQP solver. We increased the maximum number of iterations for Netlib LP evaluation to $10^6$ iterations.

While the vector RLQP policy accelerates Netlib LP optimization overall, it can slow convergence for some problems. In Figure 4 displays per-problem speedups of RLQP over OSQP. RLQP achieves speedups of up to 73x, but degrades performance for a minority of problems. We include detailed per-problem results containing solver runtime in Section E. As we evaluate the policy at fixed intervals, the solver must re-factorize the problem due to a change in $\rho$. However, the policy may update $\rho$ more times than is needed which can slow convergence for some fast well-conditioned problems. Our work is a good starting place for further research into learning methods for first-order optimization. We are extending the RLQP framework to support dynamic policy evaluation which would improve performance for these small-scale problems.

# D Maros and Mészáros Results

As with the Netlib linear problems, we evaluate the policy trained on the benchmark problems on all 138 Maros and Mészáros [5] QP problems and present the results here. We have made no effort to ensure that training problems come from the same distribution of QPs as the Maros and Mészáros problems. Many of these QPs are poorly scaled, which causes both OSQP and RLQP to sometimes fail to converge within a $600\,\text{s}$ time limit we set. Some problems that OSQP fails to solve, RLQP (vector) solves, and vice versa, while the (scalar) policy performs poorly on most of these problems (not shown). We show results for two (vector) models trained on the benchmarks. The "GNN" model includes the primal and dual residuals ($\xi_{\text{primal}}$ and $\xi_{\text{dual}}$) in $S$, while the "non-GNN" does not. In the table that follows, the bold entries are the fasted solve times in seconds and the fewest ADMM iterations, though we omit the bold when the three policies tie. We report the number of times OSQP and RLQP have the fastest solve time and fewest iterations, and observe that the difference between these indicates that time to compute the adaptation is a factor in making RLQP not outperform OSQP more often.

# E   Detailed results for Netlib LP problems

| Netlib LP Problem | $n$ | $m$ | non-zeros | OSQP | RLQP (vector) |
|---|---|---|---|---|---|
| 25FV47 | 1876 | 2697 | 12581 | **3.496** | 31.064 |
| 80BAU3B | 12061 | 14323 | 35325 | **11.569** | 52.989 |
| ADLITTLE | 138 | 194 | 562 | **0.076** | 0.079 |
| AFIRO | 51 | 78 | 153 | **0.001** | 0.002 |
| AGG2 | 758 | 1274 | 5498 | timeout | **1.183** |
| AGG3 | 758 | 1274 | 5514 | timeout | **0.415** |
| AGG | 615 | 1103 | 3477 | timeout | timeout |
| BANDM | 472 | 777 | 2966 | 0.466 | **0.264** |
| BEACONFD | 295 | 468 | 3703 | 0.025 | **0.024** |
| BLEND | 114 | 188 | 636 | 0.031 | **0.007** |
| BNL1 | 1586 | 2229 | 7118 | timeout | **0.998** |
| BNL2 | 4486 | 6810 | 19482 | **24.329** | 37.051 |
| BOEING1 | 726 | 1077 | 4553 | 3.119 | **0.348** |
| BOEING2 | 305 | 471 | 1663 | timeout | **0.198** |
| BORE3D | 334 | 567 | 1782 | 0.585 | **0.419** |
| BRANDY | 303 | 523 | 2505 | **0.548** | 0.962 |
| CAPRI | 496 | 767 | 2461 | 4.846 | **0.437** |
| CYCLE | 3378 | 5281 | 24626 | **4.931** | 29.043 |
| CZPROB | 3562 | 4491 | 14270 | 10.714 | **1.388** |
| D2Q06C | 5831 | 8002 | 38912 | **127.159** | 167.348 |
| D6CUBE | 6184 | 6599 | 43888 | 3.211 | **0.321** |
| DEGEN2 | 757 | 1201 | 4958 | **0.089** | 0.583 |
| DEGEN3 | 2604 | 4107 | 28036 | **0.730** | 3.558 |
| DFL001 | 12230 | 18301 | 47862 | **14.112** | 765.502 |
| E226 | 472 | 695 | 3240 | **0.371** | 1.126 |
| ETAMACRO | 816 | 1216 | 3353 | **0.655** | 6.718 |
| FFFFF800 | 1028 | 1552 | 7429 | timeout | timeout |
| FINNIS | 1064 | 1561 | 3824 | **2.034** | 2.657 |
| FIT1D | 1049 | 1073 | 14476 | **0.390** | 1.895 |
| FIT1P | 1677 | 2304 | 11545 | 0.478 | **0.080** |
| FIT2D | 10524 | 10549 | 139566 | **3.622** | 119.416 |
| FIT2P | 13525 | 16525 | 63809 | 0.533 | 2.332 |
| FORPLAN | 492 | 653 | 5126 | 0.061 | **0.053** |
| GANGES | 1706 | 3015 | 8643 | **4.741** | timeout |
| GFRD-PNC | 1160 | 1776 | 3605 | 0.790 | **0.288** |
| GREENBEA | 5598 | 7990 | 36668 | timeout | timeout |
| GREENBEB | 5602 | 7994 | 36677 | 122.834 | timeout |
| GROW15 | 645 | 945 | 6265 | timeout | timeout |
| GROW22 | 946 | 1386 | 9198 | **1.132** | timeout |
| GROW7 | 301 | 441 | 2913 | timeout | timeout |
| ISRAEL | 316 | 490 | 2759 | timeout | **2.781** |
| KB2 | 68 | 111 | 381 | timeout | **0.066** |
| LOTFI | 366 | 519 | 1502 | 1.599 | **0.196** |
| MAROS-R7 | 9408 | 12544 | 154256 | **253.193** | timeout |
| MAROS | 1966 | 2812 | 12103 | timeout | timeout |
| MODSZK1 | 1622 | 2309 | 4792 | **1.588** | 5.152 |
| NESM | 3105 | 3767 | 16575 | **0.811** | timeout |
| PEROLD | 1594 | 2219 | 8911 | timeout | timeout |
| PILOT-JA | 2355 | 3295 | 18571 | timeout | timeout |
| PILOT-WE | 3008 | 3730 | 12809 | timeout | timeout |
| PILOT4 | 1211 | 1621 | 8553 | timeout | timeout |
| PILOT87 | 6680 | 8710 | 81629 | timeout | timeout |
| PILOTNOV | 2446 | 3421 | 15777 | timeout | timeout |
| PILOT | 4860 | 6301 | 49235 | timeout | timeout |
| QAP12 | 8856 | 12048 | 47160 | **9.819** | 26.535 |
| QAP15 | 22275 | 28605 | 117225 | **91.608** | 137.196 |
| QAP8 | 1632 | 2544 | 8928 | 0.386 | **0.177** |
| RECIPELP | 204 | 295 | 891 | **0.002** | 0.003 |
| SC105 | 163 | 268 | 503 | **0.011** | 0.014 |
| SC205 | 317 | 522 | 982 | timeout | **0.022** |
| SC50A | 78 | 128 | 238 | **0.003** | 0.009 |
| SC50B | 78 | 128 | 226 | **0.005** | 0.023 |
| SCAGR25 | 671 | 1142 | 2396 | **0.122** | timeout |
| SCAGR7 | 185 | 314 | 650 | **0.081** | 0.087 |
| SCFXM1 | 600 | 930 | 3332 | **2.895** | timeout |
| SCFXM2 | 1200 | 1860 | 6669 | timeout | timeout |
| SCFXM3 | 1800 | 2790 | 10006 | **15.458** | timeout |
| SCORPION | 466 | 854 | 2000 | timeout | timeout |
| SCRS8 | 1275 | 1765 | 4563 | **1.156** | 7.543 |
| SCSD1 | 760 | 837 | 3148 | 0.021 | **0.008** |
| SCSD6 | 1350 | 1497 | 5666 | 0.262 | **0.017** |
| SCSD8 | 2750 | 3147 | 11334 | 0.187 | **0.031** |

| Netlib LP Problem | $n$ | $m$ | non-zeros | OSQP | RLQP (vector) |
|---|---|---|---|---|---|
| SCTAP1 | 660 | 960 | 2532 | 1.492 | **0.014** |
| SCTAP2 | 2500 | 3590 | 9834 | 1.094 | **0.056** |
| SCTAP3 | 3340 | 4820 | 13074 | 1.192 | **0.054** |
| SEBA | 1036 | 1551 | 5396 | 1.022 | **0.939** |
| SHARE1B | 253 | 370 | 1432 | **1.574** | 3.544 |
| SHARE2B | 162 | 258 | 939 | timeout | **0.030** |
| SHELL | 1777 | 2313 | 5335 | 3.615 | **0.192** |
| SHIP04L | 2166 | 2568 | 8546 | 0.716 | **0.397** |
| SHIP04S | 1506 | 1908 | 5906 | **0.091** | 0.730 |
| SHIP08L | 4363 | 5141 | 17245 | **0.372** | 0.608 |
| SHIP08S | 2467 | 3245 | 9661 | timeout | **1.034** |
| SHIP12L | 5533 | 6684 | 21809 | 5.992 | **5.682** |
| SHIP12S | 2869 | 4020 | 11153 | **1.081** | 1.874 |
| SIERRA | 2735 | 3962 | 10736 | 5.383 | **3.165** |
| STAIR | 620 | 976 | 4641 | **1.417** | timeout |
| STANDATA | 1274 | 1633 | 4504 | timeout | **0.075** |
| STANDGUB | 1383 | 1744 | 4722 | timeout | **0.079** |
| STANDMPS | 1274 | 1741 | 5152 | 1.329 | **0.028** |
| STOCFOR1 | 165 | 282 | 666 | timeout | **0.013** |
| STOCFOR2 | 3045 | 5202 | 12402 | **2.599** | 7.081 |
| STOCFOR3 | 23541 | 40216 | 100014 | timeout | timeout |
| TRUSS | 8806 | 9806 | 36642 | 10.070 | **0.770** |
| VTP-BASE | 347 | 545 | 1399 | timeout | **2.344** |
| WOOD1P | 2595 | 2839 | 72811 | timeout | **0.162** |
| WOODW | 8418 | 9516 | 45905 | **9.310** | 10.675 |
| **Total Solved:** | | | | 67 | 72 |

# F  Detailed results for Maros & Mészáros problems

| Maros & Mészáros Problem | $n$ | $m$ | non-zeros | Solve Time | | | Iteration | | |
|---|---|---|---|---|---|---|---|---|---|
| | | | | OSQP | RLQP non-GNN | RLQP GNN | OSQP | RLQP non-GNN | RLQP GNN |
| AUG2D | 20200 | 30200 | 80000 | **0.155** | 0.164 | 0.163 | 200 | 200 | 200 |
| AUG2DC | 20200 | 30200 | 80400 | **0.153** | 0.188 | 0.155 | 200 | 200 | 200 |
| AUG2DCQP | 20200 | 30200 | 80400 | 1.562 | 23.198 | **0.939** | 2200 | 26800 | **1000** |
| AUG2DQP | 20200 | 30200 | 80000 | 1.683 | 8.923 | **0.854** | 2400 | 10600 | **1000** |
| AUG3D | 3873 | 4873 | 13092 | **0.028** | 0.039 | 0.037 | 200 | 200 | 200 |
| AUG3DC | 3873 | 4873 | 14292 | **0.026** | 0.031 | 0.035 | 200 | 200 | 200 |
| AUG3DCQP | 3873 | 4873 | 14292 | **0.056** | 0.063 | 0.065 | 400 | 400 | 400 |
| AUG3DQP | 3873 | 4873 | 13092 | **0.053** | 0.064 | 0.065 | 400 | 400 | 400 |
| BOYD1 | 93261 | 93279 | 745507 | 286.552 | **275.054** | timeout | 66000 | **61400** | timeout |
| BOYD2 | 93263 | 279794 | 517049 | timeout | timeout | timeout | timeout | timeout | timeout |
| CONT-050 | 2597 | 4998 | 17199 | 0.395 | **0.237** | 17.030 | 1600 | **800** | 54800 |
| CONT-100 | 10197 | 19998 | 69399 | 12.062 | **1.766** | timeout | 8200 | **1000** | timeout |
| CONT-101 | 10197 | 20295 | 62496 | 20.508 | **3.089** | timeout | 12800 | **1800** | timeout |
| CONT-200 | 40397 | 79998 | 278799 | 352.981 | **87.121** | timeout | 33000 | **7200** | timeout |
| CONT-201 | 40397 | 80595 | 249996 | timeout | timeout | timeout | timeout | timeout | timeout |
| CONT-300 | 90597 | 180895 | 562496 | timeout | timeout | timeout | timeout | timeout | timeout |
| CVXQP1_L | 10000 | 15000 | 94966 | 84.758 | **31.133** | 104.432 | 9800 | **1800** | 6200 |
| CVXQP1_M | 1000 | 1500 | 9466 | 0.161 | **0.140** | 0.227 | 1200 | **800** | 1400 |
| CVXQP1_S | 100 | 150 | 920 | 0.004 | **0.003** | 0.035 | 800 | **600** | 6800 |
| CVXQP2_L | 10000 | 12500 | 87467 | 7.049 | 4.865 | **4.748** | 800 | **400** | **400** |
| CVXQP2_M | 1000 | 1250 | 8717 | **0.046** | 0.055 | 0.053 | 400 | 400 | 400 |
| CVXQP2_S | 100 | 125 | 846 | 0.001 | 0.001 | 0.001 | 200 | 200 | 200 |
| CVXQP3_L | 10000 | 17500 | 102465 | 99.156 | **19.785** | 23.884 | 10200 | **1000** | 1200 |
| CVXQP3_M | 1000 | 1750 | 10215 | 0.795 | **0.444** | 40.058 | 5400 | **2200** | 206400 |
| CVXQP3_S | 100 | 175 | 994 | **0.002** | 0.002 | 0.014 | 400 | 400 | 2200 |
| DPKLO1 | 133 | 210 | 1785 | 0.002 | 0.002 | 0.003 | 200 | 200 | 200 |
| DTOC3 | 14999 | 24997 | 64989 | 1.389 | **0.191** | 7.221 | 3800 | **400** | 16600 |
| DUAL1 | 85 | 86 | 7201 | 0.002 | 0.002 | 0.002 | 200 | 200 | 200 |
| DUAL2 | 96 | 97 | 9112 | 0.002 | 0.002 | 0.003 | 200 | 200 | 200 |
| DUAL3 | 111 | 112 | 12327 | 0.003 | 0.003 | 0.004 | 200 | 200 | 200 |
| DUAL4 | 75 | 76 | 5673 | 0.001 | 0.001 | 0.002 | 200 | 200 | 200 |
| DUALC1 | 9 | 224 | 2025 | 0.002 | 0.002 | 0.002 | 600 | **400** | **400** |
| DUALC2 | 7 | 236 | 1659 | 0.001 | 0.002 | 0.002 | 400 | 400 | 400 |
| DUALC5 | 8 | 286 | 2296 | 0.001 | 0.001 | 0.001 | 200 | 200 | 200 |
| DUALC8 | 8 | 511 | 4096 | **0.002** | **0.002** | 0.003 | 200 | 200 | 200 |
| EXDATA | 3000 | 6001 | 2260500 | **4.820** | 13.794 | 8.030 | **2000** | 3200 | **2000** |
| GENHS28 | 10 | 18 | 62 | 0.000 | 0.000 | 0.000 | 200 | 200 | 200 |
| GOULDQP2 | 699 | 1048 | 2791 | 0.020 | **0.008** | 0.023 | 1400 | **400** | 1200 |
| GOULDQP3 | 699 | 1048 | 3838 | 0.003 | 0.004 | 0.004 | 200 | 200 | 200 |
| HS118 | 15 | 32 | 69 | 0.000 | 0.000 | 0.000 | 800 | **400** | **400** |
| HS21 | 2 | 3 | 6 | 0.000 | 0.000 | 0.000 | 200 | 200 | 200 |
| HS268 | 5 | 10 | 55 | 0.000 | 0.000 | 0.000 | 400 | 400 | 400 |
| HS35 | 3 | 4 | 13 | 0.000 | 0.000 | 0.000 | 200 | 200 | 200 |
| HS35MOD | 3 | 4 | 13 | 0.000 | 0.000 | 0.000 | 200 | 200 | 200 |
| HS51 | 5 | 8 | 21 | 0.000 | 0.000 | 0.000 | 200 | 200 | 200 |
| HS52 | 5 | 8 | 21 | 0.000 | 0.000 | 0.000 | 200 | 200 | 200 |
| HS53 | 5 | 8 | 21 | 0.000 | 0.000 | 0.000 | 200 | 200 | 200 |
| HS76 | 4 | 7 | 22 | 0.000 | 0.000 | 0.000 | 200 | 200 | 200 |
| HUES-MOD | 10000 | 10002 | 40000 | 0.223 | 0.174 | **0.169** | 1200 | **800** | **800** |
| HUESTIS | 10000 | 10002 | 40000 | 1.380 | **0.269** | 54.088 | 7600 | **1200** | 226600 |
| KSIP | 20 | 1021 | 19938 | 0.058 | **0.025** | 0.035 | 1800 | **600** | 800 |
| LASER | 1002 | 2002 | 9462 | **0.011** | 0.012 | 0.014 | 400 | 400 | 400 |
| LISWET1 | 10002 | 20002 | 50004 | 3.324 | 278.583 | **0.851** | 11200 | 717600 | **2400** |
| LISWET10 | 10002 | 20002 | 50004 | 2.388 | 0.615 | **0.312** | 8200 | 1600 | **800** |
| LISWET11 | 10002 | 20002 | 50004 | 2.441 | 0.628 | **0.334** | 8400 | 1600 | **800** |
| LISWET12 | 10002 | 20002 | 50004 | 2.405 | 0.684 | **0.313** | 8400 | 1600 | **800** |
| LISWET2 | 10002 | 20002 | 50004 | 2.012 | 0.717 | **0.283** | 6800 | 1800 | **800** |
| LISWET3 | 10002 | 20002 | 50004 | 1.935 | 0.731 | **0.283** | 6800 | 1800 | **800** |
| LISWET4 | 10002 | 20002 | 50004 | 2.089 | 0.635 | **0.307** | 6800 | 1800 | **800** |
| LISWET5 | 10002 | 20002 | 50004 | 0.907 | 0.397 | **0.212** | 3200 | 1000 | **600** |
| LISWET6 | 10002 | 20002 | 50004 | 2.417 | 0.639 | **0.275** | 8400 | 1600 | **800** |
| LISWET7 | 10002 | 20002 | 50004 | 2.085 | 0.885 | **0.351** | 7200 | 2200 | **1000** |
| LISWET8 | 10002 | 20002 | 50004 | 2.081 | 0.791 | **0.360** | 7200 | 2200 | **1000** |
| LISWET9 | 10002 | 20002 | 50004 | 2.120 | 0.787 | **0.414** | 7200 | 2200 | **1000** |
| LOTSCHD | 12 | 19 | 72 | 0.000 | 0.000 | 0.000 | 400 | 400 | 400 |
| MOSARQP1 | 2500 | 3200 | 8512 | **0.028** | 0.046 | 0.034 | **400** | 600 | **400** |
| MOSARQP2 | 900 | 1500 | 4820 | 0.010 | 0.010 | 0.011 | 200 | 200 | 200 |
| POWELL20 | 10000 | 20000 | 40000 | 136.363 | 283.350 | **0.796** | 462400 | 653200 | **1200** |
| PRIMAL1 | 325 | 410 | 6464 | 0.005 | 0.006 | 0.006 | 200 | 200 | 200 |
| PRIMAL2 | 649 | 745 | 9339 | **0.008** | 0.011 | **0.008** | 200 | 200 | 200 |
| PRIMAL3 | 745 | 856 | 23036 | **0.020** | 0.026 | 0.021 | 200 | 200 | 200 |

| Maros & Mészáros Problem | $n$ | $m$ | non-zeros | Solve Time | | | Iteration | | |
|---|---|---|---|---|---|---|---|---|---|
| | | | | OSQP | RLQP non-GNN | RLQP GNN | OSQP | RLQP non-GNN | RLQP GNN |
| PRIMAL4 | 1489 | 1564 | 19008 | **0.019** | 0.022 | 0.020 | 200 | 200 | 200 |
| PRIMALC1 | 230 | 239 | 2529 | timeout | 0.945 | **0.006** | timeout | 94400 | **600** |
| PRIMALC2 | 231 | 238 | 2078 | timeout | 0.389 | **0.005** | timeout | 45800 | **600** |
| PRIMALC5 | 287 | 295 | 2869 | timeout | **0.005** | 0.004 | timeout | **400** | 400 |
| PRIMALC8 | 520 | 528 | 5199 | timeout | 0.435 | **0.018** | timeout | 21800 | **800** |
| Q25FV47 | 1571 | 2391 | 130523 | **6.124** | timeout | 8.155 | **27600** | timeout | 28200 |
| QADLITTL | 97 | 153 | 637 | 0.004 | 0.004 | 0.004 | 1200 | **1000** | 1000 |
| QAFIRO | 32 | 59 | 124 | 0.000 | 0.000 | 0.000 | 200 | 200 | 200 |
| QBANDM | 472 | 777 | 3023 | 0.228 | **0.044** | 0.049 | 13600 | **2000** | 2200 |
| QBEACONF | 262 | 435 | 3673 | 0.032 | **0.010** | 0.018 | 2600 | **600** | 1000 |
| QBORE3D | 315 | 548 | 1872 | 1.302 | **0.033** | 0.368 | 126200 | **2600** | 29000 |
| QBRANDY | 249 | 469 | 2511 | 0.170 | 0.090 | **0.015** | 14600 | 5600 | **1000** |
| QCAPRI | 353 | 624 | 3852 | 2.041 | 418.003 | **0.088** | 146600 | 22029400 | **4800** |
| QE226 | 282 | 505 | 4721 | 0.557 | 0.147 | **0.077** | 36400 | 7400 | **3400** |
| QETAMACR | 688 | 1088 | 11613 | 0.916 | **0.140** | 0.207 | 10000 | **1200** | 1800 |
| QFFFFF80 | 854 | 1378 | 10635 | **0.362** | 74.270 | 15.281 | **6200** | 1031600 | 201400 |
| QFORPLAN | 421 | 582 | 6112 | **0.009** | timeout | 3.255 | **400** | timeout | 153200 |
| QGFRDXPN | 1092 | 1708 | 3739 | 0.898 | **0.167** | timeout | 43400 | **6600** | timeout |
| QGROW15 | 645 | 945 | 7227 | 463.025 | timeout | **0.121** | 15832000 | timeout | **3400** |
| QGROW22 | 946 | 1386 | 10837 | 29.204 | timeout | **0.116** | 659400 | timeout | **2200** |
| QGROW7 | 301 | 441 | 3597 | 0.536 | **0.036** | timeout | 40600 | **2000** | timeout |
| QISRAEL | 142 | 316 | 3765 | 0.043 | **0.037** | 0.075 | 4800 | **3000** | 6000 |
| QPCBLEND | 83 | 157 | 657 | **0.003** | **0.003** | 0.004 | 1000 | **600** | 800 |
| QPCBOEI1 | 384 | 735 | 4253 | 0.139 | 0.058 | **0.056** | 7000 | 2200 | **1800** |
| QPCBOEI2 | 143 | 309 | 1482 | 0.908 | **0.022** | 0.028 | 148000 | **2200** | 3200 |
| QPCSTAIR | 467 | 823 | 4790 | **0.086** | 29.648 | 0.122 | **3400** | 965200 | 3800 |
| QPILOTNO | 2172 | 3147 | 16105 | **60.362** | timeout | timeout | **411200** | timeout | timeout |
| QPTEST | 2 | 4 | 10 | 0.000 | 0.000 | 0.000 | 200 | 200 | 200 |
| QRECIPE | 180 | 271 | 923 | **0.003** | 0.004 | 0.004 | 600 | 600 | 600 |
| QSC205 | 203 | 408 | 785 | 0.001 | 0.002 | 0.001 | 200 | 200 | 200 |
| QSCAGR25 | 500 | 971 | 2282 | **0.102** | timeout | 0.154 | **8800** | timeout | 9000 |
| QSCAGR7 | 140 | 269 | 602 | 0.036 | 0.435 | **0.005** | 11200 | 86400 | **1000** |
| QSCFXM1 | 457 | 787 | 4456 | 0.278 | 131.058 | 0.872 | **16400** | 5741800 | 41000 |
| QSCFXM2 | 914 | 1574 | 8285 | **1.160** | timeout | 11.558 | **32200** | timeout | 256600 |
| QSCFXM3 | 1371 | 2361 | 11501 | **1.698** | timeout | 2.708 | **30200** | timeout | 40200 |
| QSCORPIO | 358 | 746 | 1842 | timeout | 0.505 | **0.237** | timeout | 40000 | **19400** |
| QSCRS8 | 1169 | 1659 | 4560 | 0.508 | 0.084 | **0.069** | 18200 | 2400 | **2000** |
| QSCSD1 | 760 | 837 | 4584 | 0.023 | 0.017 | **0.013** | 1400 | 800 | **600** |
| QSCSD6 | 1350 | 1497 | 8378 | 0.482 | 0.035 | **0.031** | 16400 | 1000 | **800** |
| QSCSD8 | 2750 | 3147 | 16214 | 0.072 | 0.062 | **0.049** | 1200 | 800 | **600** |
| QSCTAP1 | 480 | 780 | 2442 | timeout | **0.016** | 0.117 | timeout | **1000** | 7600 |
| QSCTAP2 | 1880 | 2970 | 10007 | 0.467 | 0.060 | **0.047** | 8000 | 800 | **600** |
| QSCTAP3 | 2480 | 3960 | 13262 | 0.226 | **0.042** | 0.057 | 2800 | **400** | 600 |
| QSEBA | 1028 | 1543 | 6576 | 0.201 | timeout | **0.151** | 9400 | timeout | **5800** |
| QSHARE1B | 225 | 342 | 1436 | 0.205 | 0.419 | **0.060** | 33800 | 48400 | **6800** |
| QSHARE2B | 79 | 175 | 873 | 0.117 | 1.074 | **0.010** | 36600 | 210800 | **2000** |
| QSHELL | 1775 | 2311 | 74506 | **0.328** | 0.706 | 6.876 | **2600** | 4800 | 41200 |
| QSHIP04L | 2118 | 2520 | 8548 | 0.071 | 0.059 | **0.031** | 1800 | 1200 | **600** |
| QSHIP04S | 1458 | 1860 | 5908 | 0.039 | 0.028 | **0.024** | 1400 | 800 | **600** |
| QSHIP08L | 4283 | 5061 | 86075 | **0.192** | 0.326 | 0.253 | **600** | 800 | 600 |
| QSHIP08S | 2387 | 3165 | 32317 | 0.232 | 0.093 | **0.080** | 2400 | 800 | **600** |
| QSHIP12L | 5427 | 6578 | 144030 | 1.001 | 0.525 | **0.404** | 2000 | 800 | **600** |
| QSHIP12S | 2763 | 3914 | 44705 | 0.186 | **0.056** | 0.093 | 1600 | **400** | 600 |
| QSIERRA | 2036 | 3263 | 9582 | **0.115** | 0.179 | 0.351 | **2000** | 2400 | 4800 |
| QSTAIR | 467 | 823 | 6293 | 2.567 | 317.286 | **0.303** | 89000 | 9359600 | **8200** |
| QSTANDAT | 1075 | 1434 | 5576 | 0.245 | timeout | **0.022** | 10800 | timeout | **800** |
| S268 | 5 | 10 | 55 | 0.000 | 0.000 | 0.000 | 400 | 400 | 400 |
| STADAT1 | 2001 | 6000 | 13998 | timeout | **0.611** | timeout | timeout | **7000** | timeout |
| STADAT2 | 2001 | 6000 | 13998 | timeout | **0.244** | 10.190 | timeout | **3000** | 107800 |
| STADAT3 | 4001 | 12000 | 27998 | timeout | **1.309** | 292.029 | timeout | **7200** | 1489600 |
| STCQP1 | 4097 | 6149 | 66544 | **0.052** | 0.058 | 0.060 | 200 | 200 | 200 |
| STCQP2 | 4097 | 6149 | 66544 | 0.092 | **0.086** | 0.093 | 200 | 200 | 200 |
| TAME | 2 | 3 | 8 | 0.000 | 0.000 | 0.000 | 200 | 200 | 200 |
| UBH1 | 18009 | 30009 | 72012 | 1.106 | **0.463** | 0.711 | 2600 | **800** | 1200 |
| VALUES | 202 | 203 | 7846 | 0.008 | **0.006** | 0.010 | 600 | **600** | 1000 |
| YAO | 2002 | 4002 | 10004 | 224.794 | 7.161 | **4.181** | 4164000 | 111800 | **68000** |
| ZECEVIC2 | 2 | 4 | 7 | 0.000 | 0.000 | 0.000 | 200 | 200 | 200 |
| **Problems solved with fewest iterations:** | | | | | | | 15 | 38 | 50 |
| **Problems solved with fastest solve time:** | | | | 31 | 35 | 45 | | | |
| **Total solved before timeout:** | | | | 126 | 125 | 127 | | | |

Table 1: Detailed results for the Maros & Mészáros problems [5].