# OpenReview forum: "Accelerating Quadratic Optimization with Reinforcement Learning"
_NeurIPS.cc/2021/Conference — NeurIPS 2021 Poster_

### Official Review · Reviewer_Ktc7 · 2021-07-06

**Rating:** 7
**Confidence:** 3

**Summary:**

This paper presents a method to improve QP solving speed via reinforcement learning.  A regularization parameter used in OSQP solver is being updated via a RL policy. The approach is shown to improve OSQP performance on various benchmarks.

**Ethical Concerns:**

None.

**Limitations And Societal Impact:**

Yes, limitations are discussed.

**Main Review:**

Pro:

1. using RL to adapt QP solver parameters is novel to me.

2. approach seems to be effective across various benchmarks.

3. detailed ablation on generalization to problem sizes, problem classes and warm start problems.

4. code provided.

Con:

1. The choice of state space might need more discussion, more on this in questions.


Questions:

1. The state spaces of the RLQP(scalar) and RLQP(vector) are very different, as RLQP(scalar) only has the primal/dual residual while RLQP(vector) has other stuff.  If one provide the additional information of RLQP(vector) to RLQP(scalar), will performance of RLQP(scalar) improve?

2. Why not just use a big policy that takes all the state and output all the action for RLQP? My understanding is that the setup in the current paper allows for solving QP with different sizes, but suppose I only want to solve problem of particular size, will the "big policy" approach generate more efficient policy?

**Time Spent Reviewing:**

4

---

> ### Author Response · Authors · 2021-08-10
> **Thank you for taking the time to do this review and provide detailed feedback, reviewer Ktc7**
>
> Thank you for taking the time to do this review and provide detailed feedback.
>
> ### State space: adding additional states from vector policy to scalar policy
> The scalar policy’s state and action space is modeled directly after the inputs and outputs for the static heuristic currently deployed in OSQP. This policy is therefore easy to deploy as it requires no modification to OSQP.
>
> As it wasn’t clearly stated in the paper,
>
> |  | Single-agent scalar policy MDP | Single-agent vector policy MDP | Multi-agent vector policy MDP |
> |---:|---|---|---|
> | *State:* | Vector of <primal residuals, dual residuals> | Vector of <$\min (z_i - l_i, u_i - z_i)$, $z_i - (Ax)_i$, $y_i$, $\rho_i$, primal residuals, dual residuals> | Same as single-agent vector, but split into $m$ independent vectors |
> | *Action:* | Updated scalar $\rho$ to be shared by all $m$ constraints | Vector of $m$ $\rho$ values for all $m$ constraints | Same as single-agent vector, but split into $m$ independent $\rho$ scalars |
> | *Reward:* | -1 for each iteration until QP convergence, then 0 | Same as single-agent scalar | Same as single-agent scalar |
> | *Modification to OSQP required?*: | No | Yes | Yes |
>
> ### State space: predicting all vector $\rho$ values in one policy
> The “Big” policy is an interesting idea and we are considering how to integrate it.  This policy will still have a permutation and scaling problem.  But supposing there was not a permutation and scaling problem--then it would suggest that there could be one QP without permutation, and without scaling, solved repeatedly, in which case, a "Big Policy" may work well, and would likely have application in MPC and similar settings.  The main issue this would raise in our mind is that the dramatically increased dimensionality of the state and action space would potentially slow RL training significantly. We will explore more and add further discussion to an updated draft.

---

### Official Review · Reviewer_Kvzc · 2021-07-11

**Rating:** 7
**Confidence:** 2

**Summary:**

Summary

The paper uses RL methods to learn how to adapt the internal parameters of a QP solver to minimize solve time. The paper demonstrates superior performance compared to current standard solvers across a large class of QP problems.


**Limitations And Societal Impact:**

Yes, I read the checklist at the end of the pdf and agree with the authors' assessment

**Main Review:**


Pros

The motivation of applying RL to minimize solve time across a distribution of optimization problems seems natural to me, since solve time is non-differentiable with respect to the internal parameters of the solver. The paper also shows superior performance compared to standard solvers across a variety of QP problem classes, and therefore increases my confidence in the utility of the approach.

Cons

Given the successes of RL in solving problems where sample efficiency is not a concern, perhaps the results presented in the paper are not surprising. That being said, given that solving QP has many practical uses, I think this method and line of work will be very practically useful.

The paper demonstrates the benefit of using RL to reduce solve time, but if I am not mistaken, does not offer any analysis on why the updates by the RL policy to the internal parameters of the solver reduces solve time. It would be interesting if the authors can offer some analysis on this. For example, how different are the updates as computed by the RL policy compared to the heuristic update rule?

Limitations as a reviewer

I am not an expert in the area of using RL to speed up optimization, therefore I might have missed crucial background context in assessing the paper.


**Time Spent Reviewing:**

1

---

> ### Author Response · Authors · 2021-08-10
> **Thank you for taking the time to do this review and provide detailed feedback, reviewer Kvzc**
>
> Thank you for taking the time to do this review and provide detailed feedback. We are encouraged that you found our work to be practically useful with a natural motivation.
>
> ### Analysis of why RL accelerates convergence
> The value of rho during each iteration is known to affect convergence rate, however the best value to set it is unknown and an active area of research.  The original OSQP paper presents a heuristic that attempts to balance convergence of the primal and dual residuals, observing that when the primal is (much) lower than the dual, high values of rho will lower the primal residual faster, and vice versa for when the dual is much lower than the primal.
>
> We see that RL tends to learn a policy that matches this observation and is thus similar to the OSQP function, but the shape and extremum of the learned policy function is different. Vector RLQP also uses the observation that each value of rho can be set individually, whereas OSQP sets the entire vector based on the primal and dual residuals.  In this case, the RL policy can also take advantage of per-constraint state (instead of just the two residuals) giving it an additional advantage over OSQP.
>
> We agree with the reviewer that this is an important point, and we will include plots that visualize the learned functions for both scalar and vector versions and overlay OSQP’s heuristic policy.
>
> ### Results of paper are not surprising
> We agree that this is a case where RL can shine. Our method is actually surprisingly sample efficient as it only requires a few thousand training episodes (much fewer than other applications of RL). We actually were also quite surprised when RLQP was able to outperform a highly-tuned commercial solver like Gurobi. We originally expected minor performance improvements (10-20%) but did not expect the large speedups found in evaluation (up to 2-3x).

---

### Official Review · Reviewer_BuNG · 2021-07-11

**Rating:** 1
**Confidence:** 5

**Summary:**

Applied RL to the problem of 'auto' tuning of the ADMM class of optimization algorithms for QPs.

**Ethical Concerns:**

No.

**Limitations And Societal Impact:**

Yes.

**Main Review:**

The paper is rather light in terms of novelty. Indeed, the idea of using RL to tune ADMM algorithms is not new and has been introduced in [44]. The difference w.r.t. [44] is marginal, if any at all.

Besides, the paper is not well written; the convex optimization section is rather amateurish, same to be said for the RL part, etc.

As far as the general idea goes, I am not convinced that this line of work brings much to the optimization algorithmic area, since to tune the ADMM algorithm, or any algorithm, the authors are solving yet another optimization problem, much more complex, since non-convex and potentially large-scale, with many other hyper-parameters which need to be tuned !!! so what's the gain really ? do you use another RL algorithm to tune the optimization algorithm (e.g. ADAM) that is used to solve the first RL algorithm, etc. ? This line of research is merely transferring the problem of tuning one optimization algorithm to another optimization algorithm, and so, in my opinion, not useful as it is proposed in this paper.


**Time Spent Reviewing:**

24 hours

---

> ### Author Response · Authors · 2021-08-11
> **Thank you for taking the time to read our paper and for providing us with helpful feedback, reviewer BuNG**
>
> ### “The difference w.r.t. [44] is marginal, if any at all.”
> We apologize for the lack of clarity of the comparison to Wei et al. 2020, we will update the text to clarify the distinction, as RLQP and Wei et al. 2020 are quite different.
>
> Wei et al. 2020 develop a solution specific to unconstrained optimization for image recovery that does not support general convex problems and must learn a new policy for every new dimension of problem. In RLQP, we instead develop a general policy for any arbitrary constrained quadratic programs. RLQP also adapts to novel unseen families of problems at test time while also scaling to problems of arbitrary dimensions and permutations.
>
> The methodologies employed by Wei et al. 2020 and RLQP are also different--Wei et al. 2020 tune the more brittle denoising operations between ADMM steps while RLQP more flexibly adjusts $\rho$ to maintain support for constrained problems using a multi-agent single-policy RL.
>
> Wei et al. 2020 also evaluated simple baseline algorithms that are not SOTA, while RLQP strongly outperforms SOTA solvers on a broad set of challenging QPs--including OSQP and the Gurobi commercial solver that has been hand-tuned for benchmark suites.
>
> ### “[RL and] the convex optimization section is rather amateurish”
> We tried to clearly and briefly review the background material in convex optimization and RL for people less familiar with the related work, as we realize that different readers may have different backgrounds--e.g., people from the convex optimization community may appreciate the RL background, and vice versa. We are happy to integrate any specific feedback to improve an updated draft.
>
> ### “The authors are solving yet another optimization problem...with many other hyper-parameters which need to be tuned”
>
> While it is true that RLQP adds new hyperparameters, each added hyperparameter is much easier to tune and has far less impact on QP convergence speed than the one ADMM parameter RLQP adapts ($\rho$). Empirically, RLQP was remarkably robust to a wide range of hyperparameter choices; we chose one set of parameters for use across all experiments, and did not perform extensive hyperparameter tuning.
>
> ### “Merely transferring the problem of tuning one optimization algorithm to another optimization algorithm”
> This is the goal of RLQP--by transferring part of the online/in-the-wild adaptation problem to an offline optimization problem, RLQP accelerates online solve times. Speedups to online optimization have broad impact--fast online solve times for QPs are required for many applications (e.g., robotics MPC, SVM, etc...), and could enable new applications or save energy and computing resources.
>
> ADMM parameters can be optimally adapted using semi-definite programming (Giselsson and Boyd 2017), but this is intractable for practical applications as SDP is more complex than QP. RL enables solving the problem of optimizing ADMM parameters much more efficiently than SDP with the result of 2-3x faster convergence times.

---

### Official Review · Reviewer_riVf · 2021-07-17

**Rating:** 5
**Confidence:** 3

**Summary:**

This paper aims to speed up solving of QPs by using RL to learn adaptive parameters of the OSQP solver. The authors develop and implement an RL algorithm based on TD3 to obtain optimal parameters for OSQP that significantly speed up solution times for several classes of problems.

**Limitations And Societal Impact:**

As mentioned above, discussion of robustness of the resulting policy and when/why it can be expected to fail should be included.

**Main Review:**

The paper is well-motivated as quickly solving QPs is important for a variety of applications, and the approach is described clearly. However, I had difficulty understanding some of the results/experiments, and thus how much the proposed approach improves on existing work and where it falls short. In particular:

(1) In section 5.3, it’s mentioned that OSQP’s handcrafted policy beats RLQP on class-specific problems. Is this even with a class-specific policy? This seems to be concerning if RLQP cannot learn to outperform the handcrafted policy on specific problem classes.

(2) The results in Table 1 for QPLIB suggest that RLQP can improve solve times, but when it does not improve solve times, it often fails (timeout). This may suggest that RLQP learns parameters that also make the solver very fragile. This should be discussed in the Limitations section, and the RL policy may need to be trained to be more robust.

Overall the paper/idea are very interesting, but issues still exist with the results. The above mentioned issues should be clarified, and the fragility of the resulting policy seems to be a natural problem that would arise from using RL in this domain, and should be dealt with for wider applicability of the approach.

Minor points:

(1) RL is known to have significant variance. Did you try running TD3 several times, and observing the  performance across different policies? Or was the policy trained once and the results are just taken from this run?

(2) The scale on Figure 3 makes it difficult to read, especially for low problem sizes – perhaps the x-axis could be log scale?. Also, should there be a dashed green curve that is missing?

**Time Spent Reviewing:**

3

---

> ### Author Response · Authors · 2021-08-10
> **Thank you for your diligent review and insightful feedback, reviewer riVf**
>
> ### Handcrafted policy versus RLQP
> While the handcrafted policy outperformed the scalar RLQP policy, the vector RLQP policy outperformed the handcrafted one.  We stated this in Sec 5.3, but not clearly, and we will fix it.  We also wish to note that RLQP was not trained over problems from the distribution of motion planning algorithms we tested with. This result therefore demonstrates zero-shot generalization to an unseen problem family.
>
> ### RLQP robustness to timeout errors
> Table 1 contains results over the QPLIB set of evaluation problems. Problems from this family are out-of-distribution from the training set, so this benchmark measures zero-shot generalization. Moreover, QPLIB problem set contains “adversarial” problems that are poorly scaled with highly variable problem dimensions and sparsity patterns. If RLQP were trained on QPLIB-style problems, we expect the timeouts to no longer occur.
>
> We wish to emphasize that a “timeout” as noted in Table 1 does not mean divergence or a failure. Rather, this means that the policy took longer than a few seconds to converge. In this case, the RLQP algorithm will provably converge.
>
> We agree that in the scalar policy case, the learned solver occasionally timeout. However, The vector policy is more robust and only adds one additional timeout over the OSQP baseline. There is more work to do to make the learned policy not timeout for some problems but we believe the good zero-shot performance of RLQP vector policy is still impressive given the nature of training and test datasets.
>
> ### Minor concern: reporting variance across runs
> The variances across runs of RLQP are fairly minor. We trained 5 seeds for evaluating results of RLQP on the benchmark problem set (Fig. 2) but saw little to no difference between different policies. However, the composition of the training dataset can affect results; if there is significant skew in the classes of problems in the training dataset, then RLQP may perform slightly worse.
>
> We plot the distribution of results between multiple runs in Fig. 2 right. Note that RLQP has more consistent results than the static policy in OSQP.
>
> ### Minor concern: readability/missing line from Fig. 3
> Thanks for this note. We improved the clarity of this figure in an updated draft.  We include the dashed line as an ablation study to show how the training distribution affects performance when the problem class remains the same but the dimensionality increases.  Given the performance advantage of the vector policy we felt readers would be more interested in this than that of the scalar policy.

---

### Author Response · Authors · 2021-08-11
**Thank you to AC and all reviewers**

We sincerely thank the reviewers for their time and diligence in reviewing our submission.  It is clear that each reviewer spent the time to offer helpful suggestions, raise questions, and request clarifications, and this will improve our final submission, should it be accepted. We have included detailed comments for each reviewer.

---

### Author Response · Authors · 2021-08-17
**New experiment: interpreting policy predictions with new visualization**

To understand why the learned policy outperforms the baseline heuristics, we visualized the response of the policy to controlled inputs as recommended by several reviewers.

Figure URL (hosted on anonymized Github): https://neurips-rlqp-authors.github.io/neurips-rlqp-review-artifacts/experiments/visualizing_learned_policy

This figure plots a learned policy output $\rho$ with fixed $Ax = z + 1$, $\rho_{prev} = 0.1$, $l_i = -10$, and $u_i = 10$ and varying $z$ and $y$. From the plot, we see that when $z$ approaches either bound, the output $\rho$ increases.  We also observe that with higher $y$, the policy outputs a higher $\rho$. Both of these observations are consistent with the intent and observations behind OSQP's heuristic policy: a high $\rho$ is desirable when the primal residual is smaller than the dual, and a low $\rho$ is desirable when the dual is smaller than the primal.

Based on these insights, we are trying to extract a simple non-learned policy from RLQP. However, the learned policy does not appear to be a simple square root of the ratio of residuals. These results suggest the optimal policy is non-linear and problem dependent.

We also found that there was a missing script from the originally uploaded artifact. We have uploaded a copy of the script anonymously to the new Github repository (with anonymized account) at https://github.com/neurips-rlqp-authors/neurips-rlqp-review-artifacts/blob/main/code/train_benchmark_problems.sh.

---

### Decision · Program_Chairs · 2021-09-28

**Decision:**

Accept (Poster)

**Comment:**

This is a difficult one to assess. There were some strong opinions among the reviewers, and one reviewer that did seem a bit excessively harsh - perhaps expecting the paper to have the same sort of rigor one might find in a pure optimization paper, rather than an RL application paper.

Evaluated as an RL application paper, a key question is whether the results generalize enough to be interesting. The word "enough" is important because it's not reasonable to expect RL to find a silver bullet set of parameters that will make everything better. One might hope that RL could find useful parameters within related classes of problems. I do think that the authors provide some reasonable evidence to support the claim that they are making significant progress in this direction, so my overall assessment is on the positive side, albeit with some reservations that reflect that reality that we have seen decades of efforts to tweak optimization through RL (see, e.g., Justin Boyan's thesis from the late 90's) but not much practical impact despite some nice tables and graphs in papers.

**Consistency Experiment:**

NeurIPS has a long history of experimentation. In 2014, NeurIPS ran an experiment in which 10% of submissions were reviewed by two independent committees to quantify the randomness in the review process. This year, we repeated a variant of this experiment to see how the quality of the review process has changed over time.  This paper was part of the experiment and was therefore assigned to two committees (consisting of reviewers, an Area Chair, and a Senior Area Chair) that reached independent decisions.  If both committees made the same recommendation, this recommendation was followed. If a single committee recommended acceptance, the paper was accepted (with the exception of a few cases in which the other committee identified what we considered a fatal flaw, e.g., an error in a key result).

Both committees reached the same decision: **Accept (Poster)**

The other committee assigned to the paper recommended **Accept (Poster)**.  You can find the other set of reviews, along with any follow up discussion with the authors here:
https://openreview.net/forum?id=5FtUGRvwEF